# Analysis of the Emission Features in CdSe/ZnS Quantum Dot-Doped Polymer Fibers

**Xuefeng Peng [1,*], Zhijian Wu [1,2], Chen Ye [1], Yang Ding [1] and Wei Liu [1,*]**

1 College of Science & Technology, Ningbo University, Ningbo 315211, China
2 Faculty of Electrical Engineering and Computer Science, Ningbo University, Ningbo 315211, China
* Correspondence: pengxuefeng@nbu.edu.cn (X.P.); liuwei@nbu.edu.cn (W.L.)

**Abstract:** The emission features of Cdse/ZnS quantum dots doped step-index polymer optical fibers are computationally analyzed in this paper. Spontaneous emission and amplified spontaneous emission were calculated by a theoretical model based on the rate equations in terms of time, fiber length, and wavelength. All the calculated parameters are derived from experiments. Through the comparative analysis of the calculated and experimental results of spontaneous emission, we found that the pump power and overlap between the emission and absorption cross-sections may be the two main reasons for the red shift of the output spectra. When the pump power exceeds the threshold of amplified spontaneous emission, the width of the output spectra will rapidly decrease, the output wavelength will fall back toward the maximum emission cross-section, and the final output wavelength will still be affected by the doping concentration and pump power, while amplified spontaneous emission will not be generated when the total number of doped quantum dots is less than $1.27 \times 10^{12}$.

**Keywords:** Cdse/ZnS; quantum dots; polymer fiber; re-absorption effect; spontaneous emission; amplified spontaneous emission

## 1. Introduction

Owing to their size-dependent emission wavelengths, relatively high quantum yields, extremely large linear absorption cross-sections, and easy wet-chemically fabrication, semi-conductor quantum dots (QDs) have attracted research interest in fields such as photovoltaic devices [1–3], electroluminescent cells [4,5], and biolabeling [6]. Meanwhile, QDs (such as CdS, CdSe, CdSe/ZnS, PbS, PbSe, and PbTe) embedded into different matrices represent good gain characteristics in either film or fiber structures [7–10]. Compared with thin film waveguides, optical fibers have good symmetry, low transmission, and connection losses, and can be conveniently connected to existing communication networks. Therefore, increased research activities have been carried out in the field of QDs-doped fiber amplifiers and lasers in the last few years [11–14].

The emission properties of QDs depend on the size of the particle and can be tuned accurately over a wide band of wavelengths. Generally speaking, the highest gain appears in the proximity of the photoluminescence (PL) peak. However, when the emission and absorption cross-sections overlap significantly, the absorption of the fiber can have a strong influence on the output spectrum. In our previous work, we found that the PL spectra of fibers doped with CdSe/ZnS QDs will shift to longer wavelength (also named "red shift") with distance [15]. Similar phenomena were reported in other QDs [16,17], and the reason was attributed to the re-absorption caused by the overlap between the emission and absorption curves of QDs. The spectral shifts depend on the pump power, fiber length, and doping concentration. Moreover, the emission and absorption cross-sections can change significantly through the control of the QDs size, thus allowing us to change the gain to any wavelength of interest in a broad spectral range, which can be used for

making wavelength-tunable fiber lasers and amplifiers. However, this important issue was not given adequate attention, and almost all the spectral shifts reported just made a qualitative analysis and a quantitative calculation. Most calculations of the gain ignored the re-absorption phenomenon.

In this paper, we paid special attention to the influence of re-absorption caused by the overlap between the emission and absorption cros-sections and developed a theoretical model to simulate and analyze the evolution of the spontaneous emission (SE) and Amplified Spontaneous Emission (ASE) in QDs-doped polymer fibers. To validate the model, we prepared CdSe/ZnS QDs-doped polymer optical fibers (POFs) and measured the emission spectra at different experimental conditions and developed ad hoc finite difference algorithms to solve the model based on the emission and absorption cross-sections calculated from the measured absorption and PL spectra and some parameters of CdSe/ZnS QDs and fibers. The simulation results agreed well with the experimental data. This research can be a theoretical basis for the future development of wavelength-tunable QDs-doped fiber amplifiers and lasers. Moreover, the computational method developed could serve fibers with many other dopants.

## 2. Theoretical Model

CdSe/ZnS QDs have two main electronic energy states, with many vibrational energy levels within each one of them. The transitions between vibrational levels in each electronic energy state are rapid and nonradiative. Therefore, the energy diagram of CdSe/ZnS can be simplified as a two-level system, which can be seen in our previous works [18,19].

Commonly, the accurate change of the pumping light ($P_P$), emitting light ($P$), and the molecule population density in the excited state ($N_2$) with time ($t$) and position along the fiber ($z$) can be analyzed based on a set of rate equations similar to those of rare-earth-doped fiber. It is important to note that the emission spectra of CdSe/ZnS QDs-doped POFs can be turned by changing the fiber length, taking advantage of the significant overlap between the absorption and emission cross-sections of CdSe/ZnS QDs. Therefore, the independent variables have to take into account the wavelength $\lambda$, which allows us to carry out computational simulations of spectra and its evolution with fiber parameters, such as QDs-doped concentration and doped fiber length. To introduce this dependence, we divided the absorption and PL spectra into discrete subintervals centered at wavelengths $\lambda_k$. Therefore, either the absorption cross-section or the emission cross-section should be $\lambda$-dependent, as $\sigma_a(\lambda_k)$, $\sigma_e(\lambda_k)$ or $\sigma_a(\lambda_P)$, $\sigma_e(\lambda_P)$.

We assumed that the CdSe/ZnS QDs-doped POFs were end-pumped from the position $z = 0$, and that the wavelength of pump laser was $\lambda_P$. The rate equations can be expressed as follows [20,21]:

$$\begin{aligned}
\frac{\partial N_2(z,t)}{\partial t} = &-\frac{N_2(z,t)}{\tau} + \frac{\lambda_P \Gamma_P}{hcA_{core}}[\sigma_a(\lambda_P)N_1(z,t) - \sigma_e(\lambda_P)N_2(z,t)]P_P(z,t) \\
&+ \frac{\Gamma_e}{hcA_{core}} \sum_{k=1}^{K} \lambda_k[\sigma_a(\lambda_k)N_1(z,t) - \sigma_e(\lambda_k)N_2(z,t)]P(z,t,\lambda_k)
\end{aligned} \tag{1}$$

$$\begin{aligned}
\frac{\partial P(z,t,\lambda_k)}{\partial z} = &\Gamma_e[\sigma_e(\lambda_k)N_2(z,t) - \sigma_a(\lambda_k)N_1(z,t)]P(z,t,\lambda_k) - \alpha(\lambda_k)P(z,t,\lambda_k) \\
&- \frac{1}{v_e}\frac{\partial P(z,t,\lambda_k)}{\partial t} + \frac{N_2(z,t)}{\tau}h(\frac{c}{\lambda_k})\sigma_{e-sp}(\lambda_k)\beta A_{core} \quad k = 1,\ldots K
\end{aligned} \tag{2}$$

$$\frac{\partial P_P(z,t)}{\partial z} = -\Gamma_P[\sigma_a(\lambda_P)N_1(z,t) - \sigma_e(\lambda_P)N_2(z,t)]P_P(z,t) - \alpha(\lambda_P)P_P(z,t) - \frac{1}{v_P}\frac{\partial P_P(z,t)}{\partial t} \tag{3}$$

where $N_2$(z,t) is the density of CdSe/ZnS QDs in the excited states, $N_1$(z,t) is the density of CdSe/ZnS QDs in the non-excited state, and $N_1 = N - N_2$, $N$ is the total density. $P$(z,t,$\lambda$) is the resulting light power at the wavelength $\lambda_p$, $P_p$(z,t) is the pump power. $h$ is the Planck's constant, $c$ is the speed of light, and $\tau$ is the spontaneous lifetime of CdSe/ZnS QDs. $A_{core}$ is the diameter of doped fiber core, $v_z$ is the speed of light in the fiber core. $\sigma_a(\lambda_k)$ and $\sigma_e(\lambda_k)$ represent the absorption and stimulated emission cross-section at wavelength $\lambda_k$, respectively. Similarly, $\sigma_a(\lambda_p)$ and $\sigma_e(\lambda_p)$ represent the absorption and stimulated emission

cross-section at wavelength $\lambda_p$. $\sum_{\text{e-sp}}(\lambda_k)$ represents the spontaneous emission cross-section at wavelength $\lambda_k$.

The first term on the right-hand side (RHS) of Equation (1) is the spontaneous decay. The absorption and stimulated radiation caused by $P_p(z,t)$ are considered in the second term. The last term is the absorption and stimulated radiation caused by $P(z,t,\lambda)$, correspond to the re-absorption effect.

The first term on the RHS of Equation (2) represents the absorption and stimulated radiation caused by $P(z,t,\lambda)$. The second and third terms are the attenuation caused by material absorption and the propagation of the $P(z,t,\lambda)$ in the fiber core. The last term accounts for the spontaneous emission.

In Equation (3), the first term on the RHS represents the absorption and stimulated radiation of the pump light. The second term is the attenuation caused by material absorption, and the last term represents the propagation of the $P_p(z,t)$ in the fiber core.

The initial boundary conditions associated with the above partial differential equations can be written as $P_p(0) = P_0$; $P_p(0,t,\lambda_k) = 0$ (k = 1, 2, ... , K), which means that the CW pump power is $P_0$ at $t = 0$ and will be launched into the fiber at $z = 0$. The aforementioned rate equations under steady-state conditions can be numerically solved through the finite-difference method, and expressed as

$$N_2(z) = \frac{\frac{\lambda_P \Gamma_P}{hcA_{core}}\sigma_a(\lambda_P)NP_P(z) + \frac{\Gamma_e}{hcA_{core}}\sum_{k=1}^{K}\lambda_k\sigma_a(\lambda_k)NP(z,\lambda_k)}{\frac{\lambda_P \Gamma_P}{hcA_{core}}[\sigma_a(\lambda_P) + \sigma_e(\lambda_P)]P_P(z) + \frac{1}{\tau} + \frac{\Gamma_e}{hcA_{core}}\sum_{k=1}^{K}\lambda_k[\sigma_a(\lambda_k) + \sigma_e(\lambda_k)]P(z,\lambda_k)} \tag{4}$$

$$P(z+\Delta z, \lambda_k) = P(z,\lambda_k)\{\Gamma_e[-\sigma_e(\lambda_k)N + (\sigma_a(\lambda_k) + \sigma_e(\lambda_k))N_2(z)]P(z,\lambda_k) \\ - \alpha(\lambda_k)P(z,\lambda_k) + \frac{N_2(z}{\tau}h(\frac{c}{\lambda_k})\sigma_{e-sp}(\lambda_k)\beta A_{core}\}\Delta z \quad k = 1,\ldots K \tag{5}$$

$$P_P(z+\Delta z) = P_P(z) + \{-\Gamma_P[\sigma_a(\lambda_P)N - (\sigma_a(\lambda_P) + \sigma_e(\lambda_P))N_2(z)]P_P(z) - \alpha(\lambda_P)P_P(z)\}\Delta z \tag{6}$$

## 3. Model Parameters Derived Experimentally

In order to get the accuracy model parameters, we measured the absorption and emission spectra of CdSe/ZnS QDs, and optical parameters of QDs-doped POFs.

The CdSe/ZnS QDs were provided by Shanghai Institute of Technical Physics, Chinese Academy of Sciences. The 3.8 nm particle size was obtained from a transmitted electron microscopy (TEM) and calculated based on the first absorption peak of the absorption spectrum of the QDs [22]. Finally, the emission and absorption spectra of CdSe/ZnS QDs dissolved in toluene were recorded and plotted in Figure 1, showing peaks centered at 558.2 nm and 581.9 nm, respectively. CdSe/ZnS QDs-doped POFs with a 132 μm inner diameter core were fabricated based on the method reported in our previous work [15]. The fiber diameters were observed by Super Long Depth of View Optical Microscope (Keyence, VHX-100E, Osaka, Japan). The refractive indexes used for the theoretical calculation for $n_{clad}$ and $n_{core}$ were 1.493 and 1.458, respectively.

The absorption and emission cross-sections can be calculated from the absorption and PL spectra of CdSe/ZnS QDs solution based on the research of Yu [22] and Digonnet [23]. The first absorption cross-section of CdSe QDs was associated with diameter $D$, transition energy $\Delta E$, molarity $c$, and number density $n_q$, and can be calculated from an empirical equation [22]: $\sigma_a(\lambda_{\text{peak}}) = 1600\Delta E D^3 cn_q$. The absorption emission cross-section at any arbitrary wavelength could be deduced from the absorption spectra of the QDs. The stimulated emission cross-section can be obtained by solving the Mc-Cumber equation [23]:

$$\sigma_e(\lambda) = \sigma_a(\lambda)\exp[\frac{hc(\lambda_0^{-1} - \lambda^{-1})}{kT}] \tag{7}$$

where $h$ is the Planck's constant, $c$ is the speed of light, $k$ is the Boltzmann's constant, and $\lambda_0$ is the intersection of the absorption and stimulated emission spectra, which is 571.3 nm in this paper.

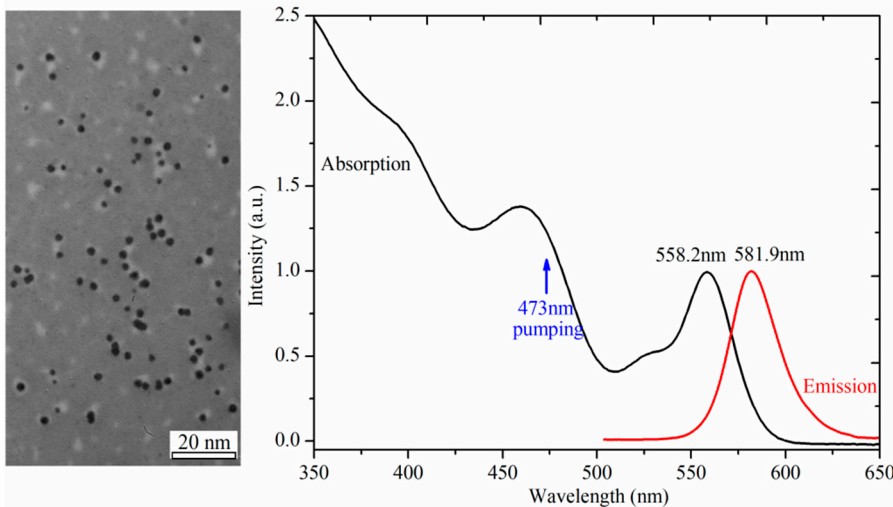

**Figure 1.** TEM image of the 3.8 nm CdSe/ZnS QDs, and the corresponding emission and absorption spectra.

The parameters used in our model are summarized in Table 1.

**Table 1.** Parameter values used in the simulation.

| Parameter | Notation | Value |
| --- | --- | --- |
| Pump wavelength | $\lambda_{peak}$ | 473 nm |
| Core radius | $r$ | 66 μm |
| Number density of QDs | $n_q$ | $1.86 \times 10^{15}$ cm$^{-3}$ |
| Absorption cross-section (558.2 nm) | $\sigma_a(\lambda_{peak})$ | $6.22 \times 10^{-22}$ m$^2$ |
| Emission cross-section (581.9 nm) | $\sigma_e(\lambda_{peak})$ | $6.55 \times 10^{-22}$ m$^2$ |
| Absorption cross-section (473 nm) | $\sigma_a(\lambda_p)$ | $7.85 \times 10^{-22}$ m$^2$ |
| Emission cross-section (473 nm) | $\sigma_e(\lambda_p)$ | 0 |
| Spontaneous lifetime | $\tau$ | 20 ns |

All the cross-sections and $n_q$ corresponding to l ppm concentration of CdSe/ZnS QDs.

The output spectra of CdSe/ZnS QDs with different fiber lengths and doped concentrations were measured experimentally. The experimental arrangement is shown in Figure 2. A 473 nm semiconductor laser (CNI, MBL-III-100 mW, Changchun, China) was used as the excitation source. After transmitting through a ×20 objective lens, the excitation beam was fed into the QDs POF and then guided into spectrometer for the spectral measurement.

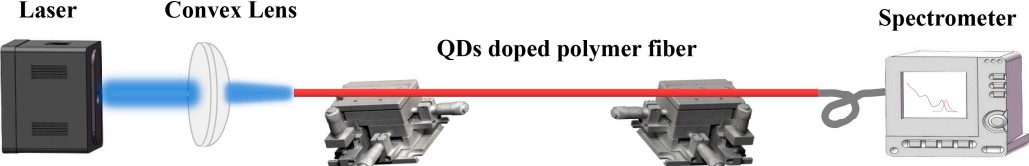

**Figure 2.** Schematic diagram of spectral measurement.

## 4. Results and Discussion

### 4.1. Analysis of the Spontaneous Emission

Spectral properties of CdSe/ZnS QDs-doped POFs were calculated and analyzed through our theoretical theory and formulas considering the influence of some important

factors. When the pump power was below the threshold of ASE, the output spectra were mainly produced by SE, and the obvious spectral red shift and width decrease caused by the re-absorption effect appeared.

Figure 3 shows the comparison of SE spectra between calculation and experiment under different fiber lengths (1–15 cm) with a QDs doping concentration of 3 ppm, fiber diameter of 100 μm, and pump power of 50 mW. The calculation results were highly consistent with the experiments. The output SE intensity first increased with the length of the doped fiber and reached the maximum at 4 cm. The short-wavelength and long-wavelength parts of the spectra were approximately symmetric, and the peak wavelength shifted slightly from 581.9 nm to 584.6 nm. It indicates that the pump power was sufficient, and the re-absorption effect was not significant. As the fiber continued to increase, the output power appeared to decrease, and the symmetry of the spectra was broken. The intensity of the short-wavelength part was significantly reduced, indicating that the pump power appeared to be insufficient, and the percentage of output light caused by re-absorption effect increased significantly. The peak wavelength increased from 584.6 nm to 594.3 nm with the increase of the fiber length from 5 to 15 cm.

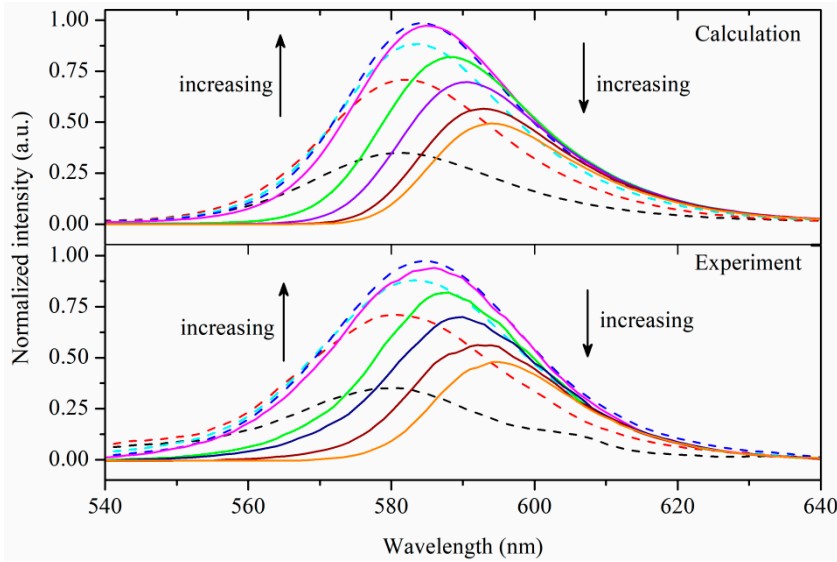

**Figure 3.** The comparison of spontaneous emission spectra between calculation and experiment under different fiber lengths (1–15 cm) in which the directions of the arrows represent the increase of the fiber length from 1 to 2, 3, 4, 5, 7, 10, 12, and 15 cm.

To further verify the correctness of our theoretical model mentioned above, CdSe/ZnS-doped fibers with concentrations of 2 ppm, 3 ppm, 4 ppm, and 5 ppm were prepared, and the peak wavelengths were measured and compared with the calculated results (shown in Figure 4). The peak wavelength variations of the output SE spectra of the four kinds of CdSe/ZnS QDs-doped fibers within the length of 1–17 cm were 582.3–592.8 nm, 583.1–600.3 nm, 583.9–606.5 nm, and 585.1–610.4 nm, with red shifts of 10.5, 17.2, 22.6, and 25.3 nm, respectively. The absolute value of red shift becomes larger as the doping concentration increases, which may be due to the fact that in the case of fixed pump power, the larger the doping concentration, the more the pump power could be absorbed, and the larger the generated SE spectrum intensity. Therefore, the re-absorption effect will be more pronounced, and the percentage of output light caused by re-absorption effect will increase, making the red shift more obvious. The values of the calculated peak wavelength red shift at the four doping concentrations were in general agreement with the experimental results, indicating the accuracy and feasibility of our theoretical model.

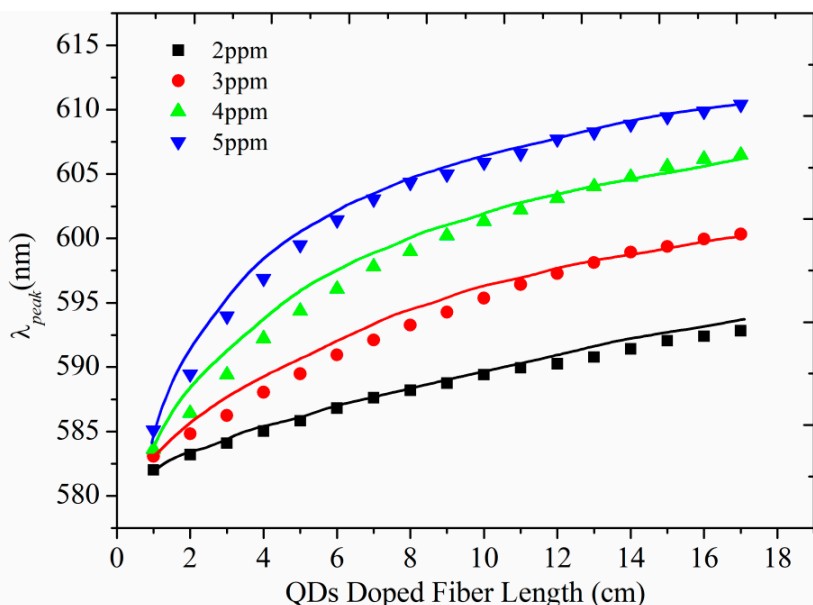

**Figure 4.** Peak wavelength of the CdSe/ZnS QD-doped POFs as a function of the doped fiber length (the solid lines were the simulation results, and the dots were the experimental results).

We can also see from Figure 4 that the peak wavelength red shifts of the four doping concentrations show an approximate linear growth in the initial fiber length. The lengths in which the slope of the red shift with fiber length remain essentially unchanged for the four doping concentrations were 17, 10, 7, and 5 cm. The larger the doped concentration, the shorter the length of the linear variation in peak wavelength. As the length continues to increase, the slope will become smaller, and the peak wavelength red shift curve will become flatter. In order to exclude the effect of PL stability on red shift, the peak wavelength and output PL intensity of 15-cm-long QDs-doped POFs with concentration of 2 ppm as a function of time were measured and shown in Figure 5. As we can see, the peak wavelength was almost equal after 60 min, and the SE intensity decreased slightly, indicating the stability of our QDs-doped POFs.

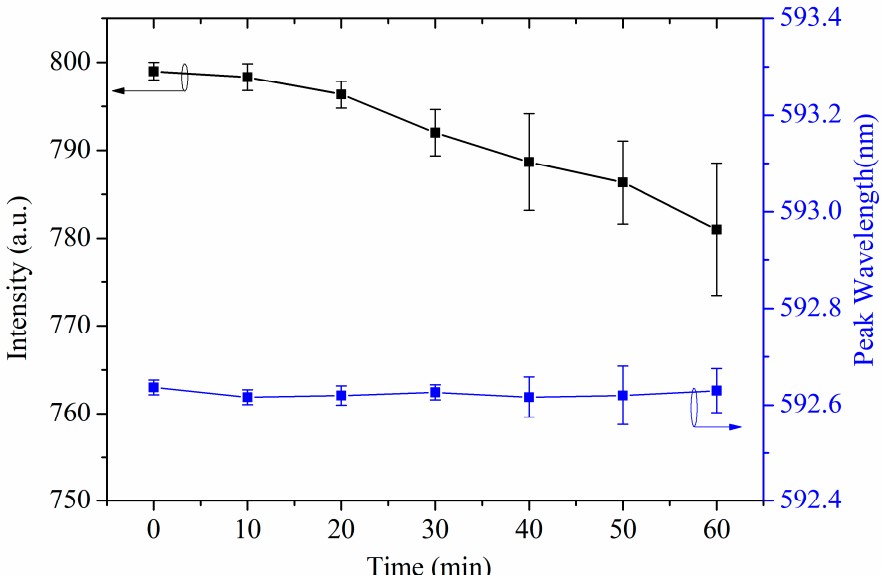

**Figure 5.** Peak wavelength and output SE intensity of a 15-cm-long POFs with concentration of 2 ppm as a function of time.

According to our analysis, there are two possible reasons: (1) it is related to the pump power; (2) it is related to the overlap of absorption and emission cross-section.

Firstly, under the same pump power, the number of QDs that can be excited was fixed, and the length of the fiber with linear increase of peak wavelength multiplied by the doping concentration was almost a constant value for the four concentrations. The higher doping concentration of QDs will result in a higher rate of pump light consumption and stronger output SE intensity in a shorter fiber length, so the re-absorption effect will be more pronounced and the red shift of peak wavelength will be larger. When doped QDs exceeds the maximum number that can be excited, the pump light will be rapidly consumed, and the pump power became very low at the back of the fiber, leading to an increase in the percentage of SE with longer wavelength generated by the re-absorption effect in the whole output spectrum. However, since the absolute intensity of SE light was much lower than that of pump, the slope of the spectral red shift due to the re-absorption effect did not increase with the fiber length but decreased gradually. To verify our assumptions, we introduced the concept of average wavelength [24], which was defined as:

$$\lambda_{av} = \frac{\int_{-\infty}^{\infty} \lambda P(\lambda) d\lambda}{\int_{-\infty}^{\infty} P(\lambda) d\lambda} \tag{8}$$

$\lambda$ was the wavelength, $P(\lambda)$ was the intensity of the light at the wavelength of $\lambda$. The discretization of $\lambda$ was carried out by dividing the full wavelength spectrum into several small areas, each one with its own step size $d\lambda$. If the re-absorption effect existed, then the SE light in the short wavelength band between $\lambda_1$ to $\lambda_1 + d\lambda$ might be re-absorbed and generated a new light with relative longer wavelength at $\lambda_2$ to $\lambda_2 + d\lambda$, which would lead to the symmetry of the whole spectrum being broken and the average wavelength will be larger than the peak wavelength. The curves of peak wavelength and average wavelength of two kinds of fiber with doping concentrations of 2 ppm and 5 ppm as a function of doped fiber length are shown in Figure 6.

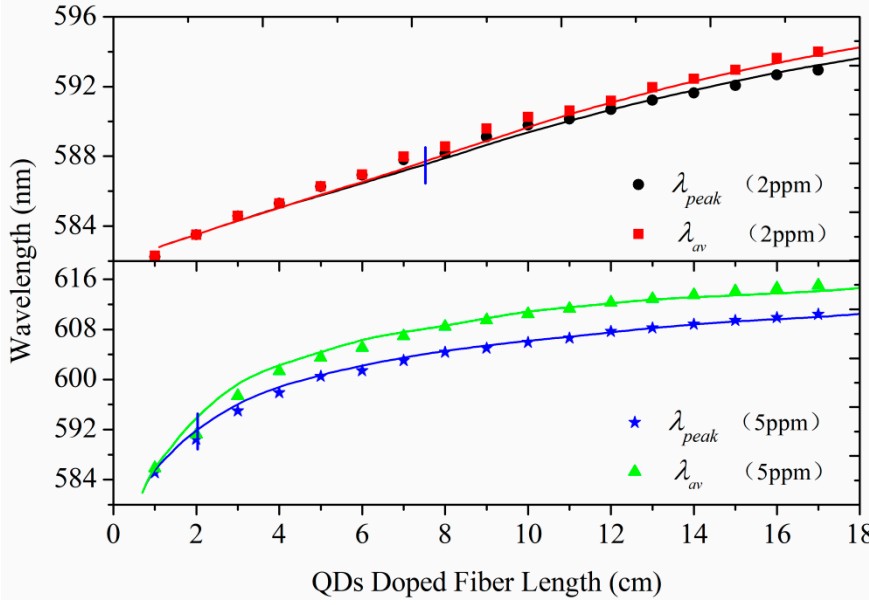

**Figure 6.** Peak wavelength and average wavelength of two kinds of fiber with doping concentrations of 2 ppm and 5 ppm as a function of doped fiber length (the solid lines were the simulation results, and the dots were the experimental results).

In the QDs-doped POF with 2 ppm concentration, the peak wavelength and average wavelength were basically consistent in the initial length of 7 cm, and the wavelength increased from 581.9 nm to 587.5 nm, with a red shift of 5.6 nm. Starting from about 7.5 cm, the peak wavelength and the average wavelength were obviously separated, and

the average wavelength became larger, indicating that the proportion of the re-absorption effect in the output SE spectrum increased significantly. In the QDs-doped POF with 5 ppm concentration, the peak wavelength and the average wavelength reached to 591.4 nm at 2 cm with a red shift of 9.5 nm, and then appeared to be clearly separated when the fiber length exceeded 3 cm. The peak and average wavelengths reached 610.4 and 615.1 nm at the end of the 17 cm fiber, respectively. The red shift was much more than that when the doping concentration was 2 ppm, indicating that the higher the doping concentration, the greater the re-absorption effect. However, it is worth noting that the difference remained roughly the same after 8 cm when the doping concentration was 5 ppm, showing that the re-absorption effect was mainly in the initial length of 8 cm. This may be related to the reduced intensity of the output light after 8 cm.

In order to verify our hypothesis, we measured and calculated the output SE light under different fiber lengths and found a good match between them, as shown in Figure 7. To increase the comparability, we normalized the output SE intensities of the four kinds of QDs-doped POFs with concentrations of 2 ppm, 3 ppm, 4 pm, and 5 ppm. The doped fiber lengths corresponding to the maximum output SE intensity were about 7.8, 6.1, 4.7, and 2.9 cm, respectively. Overall, 7.8 and 2.9 cm were roughly equivalent to the lengths (7.5 and 2 cm) that peak wavelength and average wavelength appear to be clearly separated for the fibers with concentrations of 2 ppm and 5 ppm, as shown in Figure 6. The output SE intensity reached the maximum, indicating that the pump light was basically absorbed, and the output spectrum will be mainly generated by the re-absorption effect as the fiber length continues to increase. The higher the doping concentration, the more obvious the re-absorption effect, the greater the absolute value of red shift, and the faster the output SE intensity decreases. This can also be demonstrated from the variation of the full width at half maximum (FWHM) of the four kinds of doped fiber with different concentrations in Figure 8. It can be seen that the larger the concentration, the faster the FWHM decreased, indicating that the re-absorption effect was also more obvious. This result was in general agreement with the analysis in Figures 6 and 7. The decrease rates of the FWHM became smaller in the rear part of the fibers, mainly because the absolute SE intensity became small as the fiber lengths increased. Therefore, although the proportion of re-absorption effect in the output spectrum increased, the absolute amount of red shift was still small.

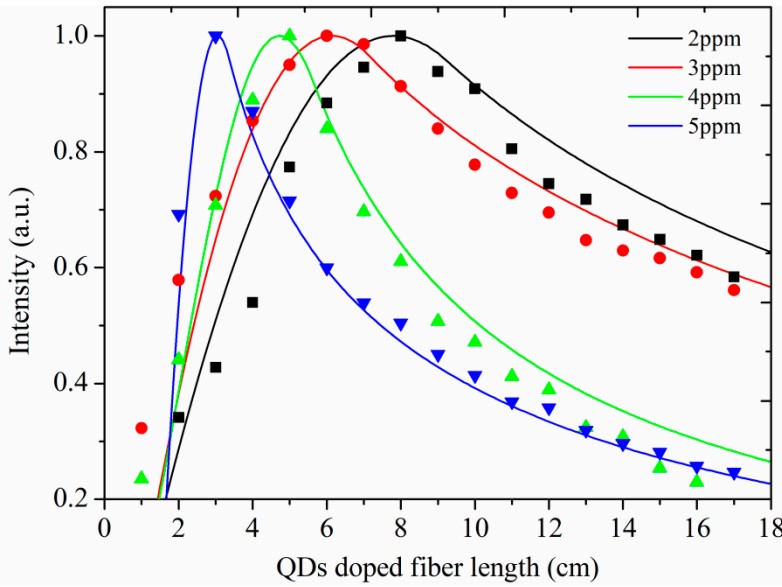

**Figure 7.** Output SE intensity of four kinds of POFs as a function of doped fiber length (the solid lines were the simulation results, and the dots were the experimental results).

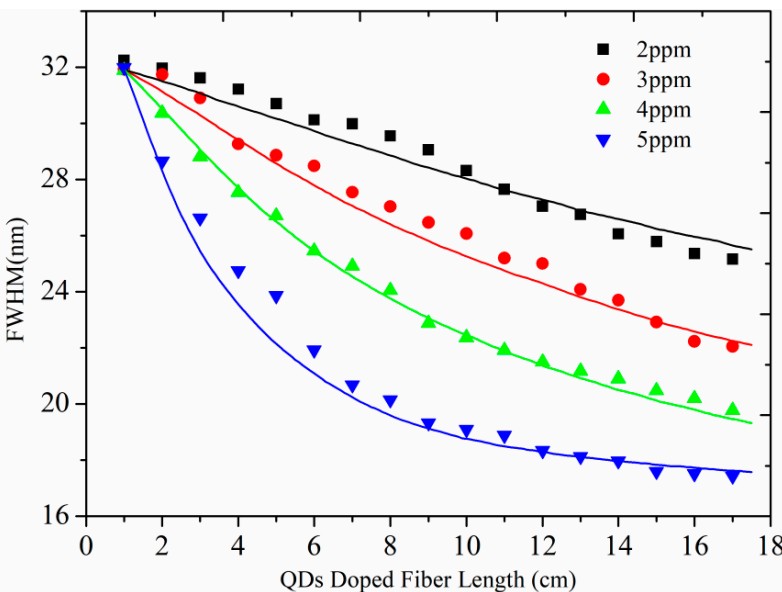

**Figure 8.** FWHM of the output SE spectra as a function of doped fiber length (the solid lines were the simulation results and the dots were the experimental results).

The slope of the red shift curve shown in Figure 4 decreases with the increase of fiber length, which may also be related to the absorption and emission cross-sections of QDS. It can be seen from Figure 9 that there were obvious overlaps between the absorption and emission cross-sections of CdSe/ZnS QDs. Since the absorption cross-sections are almost unchanged, the larger the peak wavelength of the output SE spectrum, the less overlap between the emission and absorption cross-section. In other words, the output SE spectrum that can be re-absorbed will become smaller and smaller with the increase of the peak wavelength. To verify our speculation, we introduced the concept of overlap coefficient, which was defined as:

$$\gamma_{overlap} = \frac{\int_{-\infty}^{\infty} \lambda \sigma_{overlap}(\lambda) d\lambda}{\int_{-\infty}^{\infty} \lambda \sigma_e(\lambda) d\lambda} \tag{9}$$

$\sigma_e$ and $\sigma_{overlap}$ were the emission cross-section and the emission cross-section that can be reabsorbed, respectively. We calculated the overlap coefficients of ten kinds of SE spectra with the peak wavelength from 581.9 to 626.9 nm and intervals of 5 nm, under the assumption that the SE spectra shapes were consistent. The overlapping coefficients were 0.34, 0.27, 0.21, 0.16, 0.12, 0.17, 0.078, 0.064, 0.05, and 0.05, respectively, which gradually decrease with the increase of peak wavelength. The average wavelength red shift of the output spectra at the end of the 17-cm-long fibers with the overlap coefficient for four doping concentrations of 2 ppm, 3 ppm, 4 ppm, and 5 ppm is shown in Figure 9. Under the same doping concentration, the larger the overlap coefficient, the larger the red shift. The maximum red shift under 2 ppm and 5 ppm doping concentration were 8.6 and 29.6 nm. It should be noted that as the SE wavelength becomes larger, the overlap coefficient decreases, so the maximum red shift in the experiment should be somewhat smaller than the value shown in Figure 10. The calculated red shift was smaller than that of 12.0 and 33.1 nm obtained in the experiment shown in Figure 6 because the effect of output SE intensity was not considered. The final red shift of the output spectra was the result of the combined effect of SE intensity and overlap coefficient.

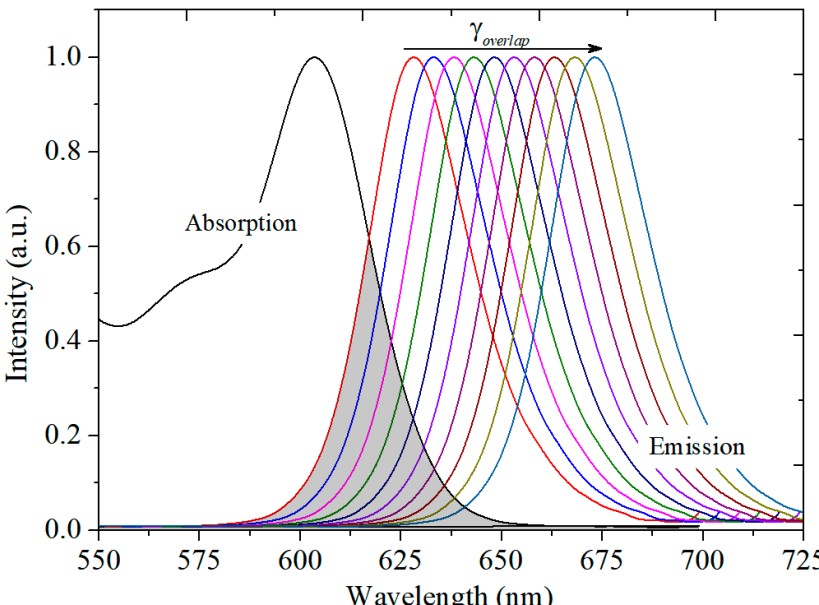

**Figure 9.** The overlap coefficients between the absorption and emission cross-sections with different peak wavelengths.

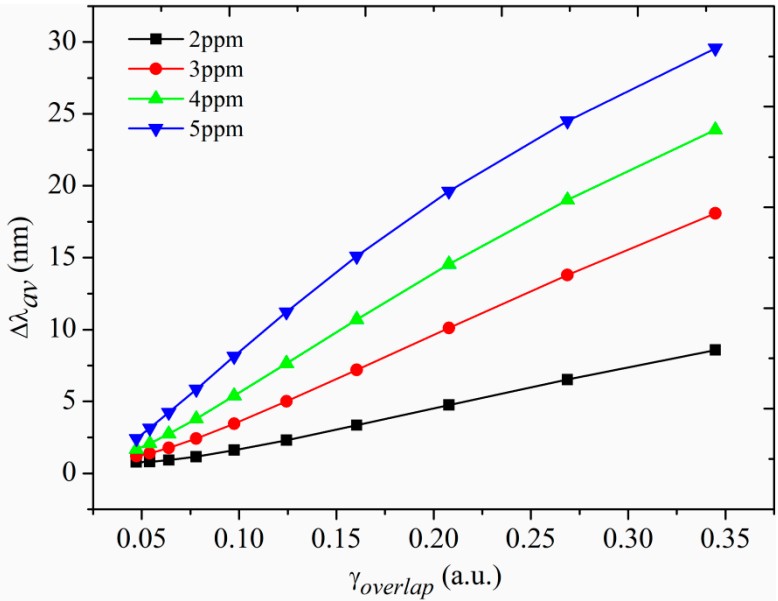

**Figure 10.** The red shift of average wavelength as a function of the overlap coefficients.

### 4.2. Analysis of the Amplified Spontaneous Emission

If the pump power is high enough, ASE phenomenon will be produced. The output light intensity at the end of the 17-cm-long fibers with concentrations of 1 ppm, 1.5 ppm, and 2 ppm were calculated and shown in Figure 11a–c. As we can see, with the increasing of the pump, the slope of the output light increased obviously and appeared a pump threshold (PT). The PTs of the three kinds of doped fiber were about 73, 102, and 127 mW, increasing with the rise of doping concentration. The FWHM and average wavelength $\lambda_{av}$ of the output spectra generated under different pump power were calculated and shown in Figure 11d,e. Special attention was paid to clarify the re-absorption effect, especially during the conversion of SE to ASE.

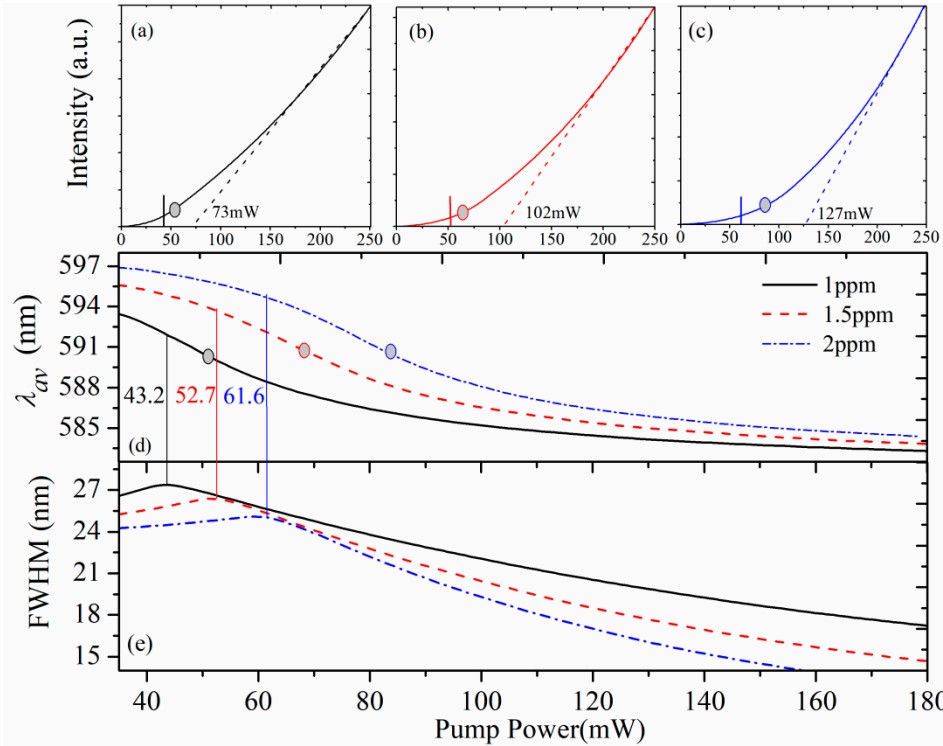

**Figure 11.** The output light intensity at the end of the 17-cm-long POFs with concentrations of (**a**) 1 ppm, (**b**) 1.5 ppm and (**c**) 2 ppm. The corresponding $\lambda_{av}$ (**d**) and FWHM (**e**) of the output light as a function of the pump power.

When the pump power was low, the output light was generated by SE. With the increase of pump, the proportion of SE that could be re-absorbed in the transmitted light decreased gradually, and the re-absorption effect weakened, leading to a decrease of $\lambda_{av}$ and a slow growth of the FWHM. The FWHM of the three kinds of doped fiber with concentrations of 1 ppm, 1.5 ppm, and 2 ppm increased from 26.62, 25.27, and 24.25 nm to their respective maximum values of 27.36 nm, 26.38 nm, and 25.21 nm. The pump powers corresponding to the three maximum FWHM were 43.2, 52.7, and 61.6 mW, and were labeled with vertical lines in Figure 11a–c. All of them were in the places where the output light intensity would increase significantly later. As we continued to increase the pump power, the output light was in a state of transition from SE to ASE, the slope of $\lambda_{av}$ became significantly larger, leading to the FWHM of the output spectra narrowing significantly, and a blue shift of the $\lambda_{av}$ toward the SE peak wavelength, where the position of the maximum emission cross-section was observed. When the pump power was larger than PTs, the number of modes decreases due to energy transfer from lower power modes to those situated near the emission peak and the ASE generates, the output intensity increases sharply and and the $\lambda_{av}$ will blue shift slightly toward the SE peak wavelength, indicating that re-absorption effect was significantly suppressed. However, the slope of FWHM was still high, and it will continue to decrease as ASE intensity increases.

The FWHM and $\lambda_{av}$ were greatly affected by the doping concentration. Under the same pump power, the larger the doping concentration, the narrower the FWHM and relatively larger $\lambda_{av}$ will be. The output ASE intensity and FWHM of many different fibers with concentrations from 0.5 ppm to 5 ppm under the pump power of 150 mW were calculated and shown in Figure 12. The output light grows rapidly with the increase of doping concentration, the ASE intensity of the 5 ppm POF was about 150 times larger than that of the 0.5 ppm POF. Correspondingly, the FWHM decreased from 20.58 nm to 7.85 nm.

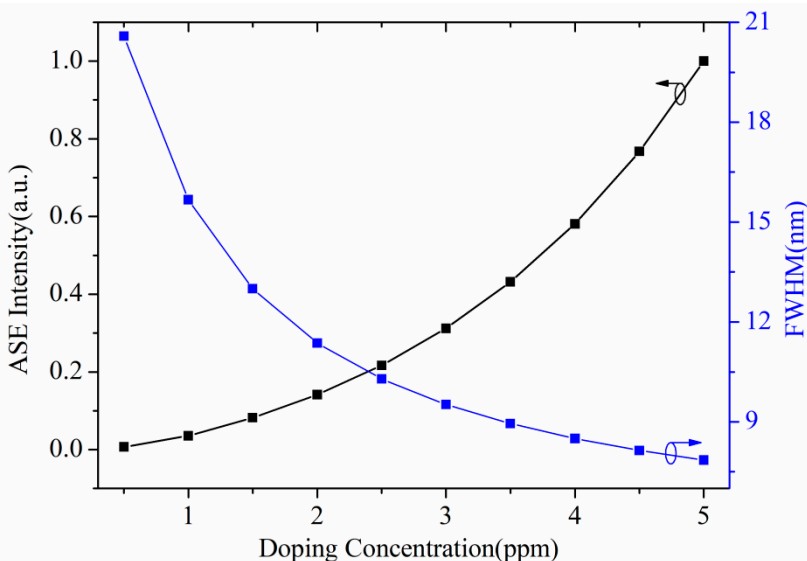

**Figure 12.** The output ASE intensity and FWHM as a function of the QDs doping concentrations.

The $\lambda_{av}$ of these POFs with concentrations from 0.5 ppm to 5 ppm under the pump power of 50 mW, 100 mW, 150 mW, and 200 mW are shown in Figure 13. The output $\lambda_{av}$ was much larger when the pump power was 50 mW, which might be attributed to the fact that the pump power was not large enough to reach the ASE PTs, so the output light was mainly generated by the SE. When the pump power up to 100 mW, 150 mW, and 200 mW, the main output light was generated by ASE, and the $\lambda_{av}$ increased almost linearly, which means that although the re-absorption effect will decrease, it will still exist and affect the $\lambda_{av}$ slightly.

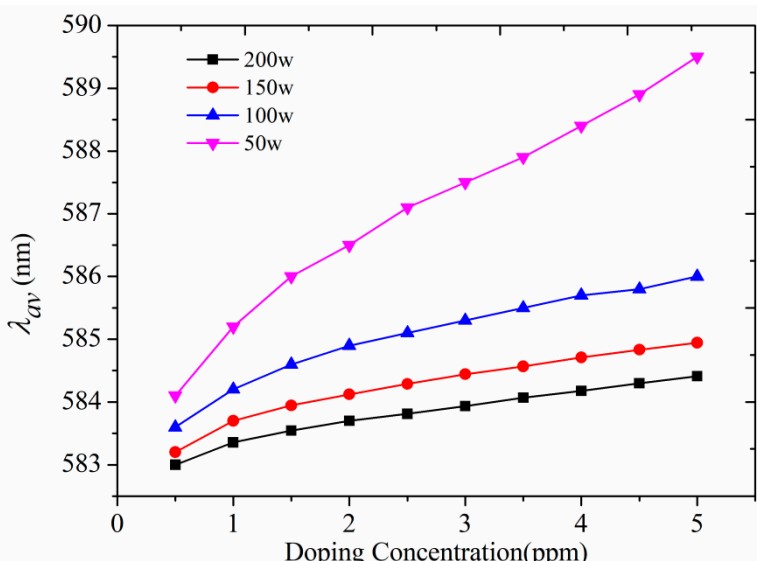

**Figure 13.** The average wavelength $\lambda_{av}$ as a function of the QDs doping concentrations.

In order to further discuss the ASE PTs generation at different doping concentrations, we calculated the emission spectra of 20 cm-long QDs-doped fibers with concentrations of 0.1 ppm–5 ppm, as shown in Figure 14. It can be seen that when the doping concentration is low enough (<0.3 ppm), ASE will not be generated. When the doping concentration is larger than 0.3 ppm, the output intensity of ASE and the ASE PT will increase with the growth of the doping concentration. The PTs increased rapidly at first, then flattened gradually, and the PT value stabilized at about 150–170 mW.

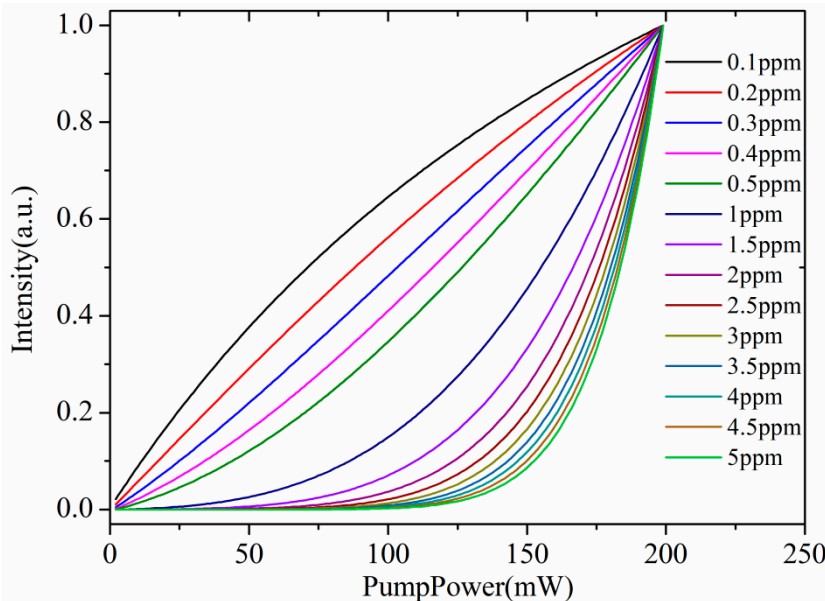

**Figure 14.** The ASE intensity of 20 cm-long QDs-doped POFs as function of pump powers with concentrations of 0.1 ppm–5 ppm.

To further clarify the influence of fiber length and doping concentration on ASE PTs, the ASE PTs of fibers with 1–20 cm fiber lengths and 0.1 ppm–10 ppm concentrations were calculated and shown in Figure 15. As we can see, PT will increase with the doping concentration and fiber length. Under the condition that with higher doping concentration and longer doped fiber length, the corresponding ASE PT will be larger, and the output power will also be higher. It is worth noting that the dark blue color in Figure 15 means that ASE will not be generated under these conditions and there will be no PTs. Although the dark blue color in Figure 15 exists either under the condition that short fiber lengths have high doping concentrations or that long fibers have low doping concentrations, the product of fiber lengths and doping concentrations was almost the same based on our analysis, and the product was about 5 ppm×cm, corresponding to a CdSe/ZnS QDs total number of $1.27 \times 10^{12}$. This means that if the total number of QDs in POFs is insufficient, ASE will not be generated, and all the output light is produced by SE.

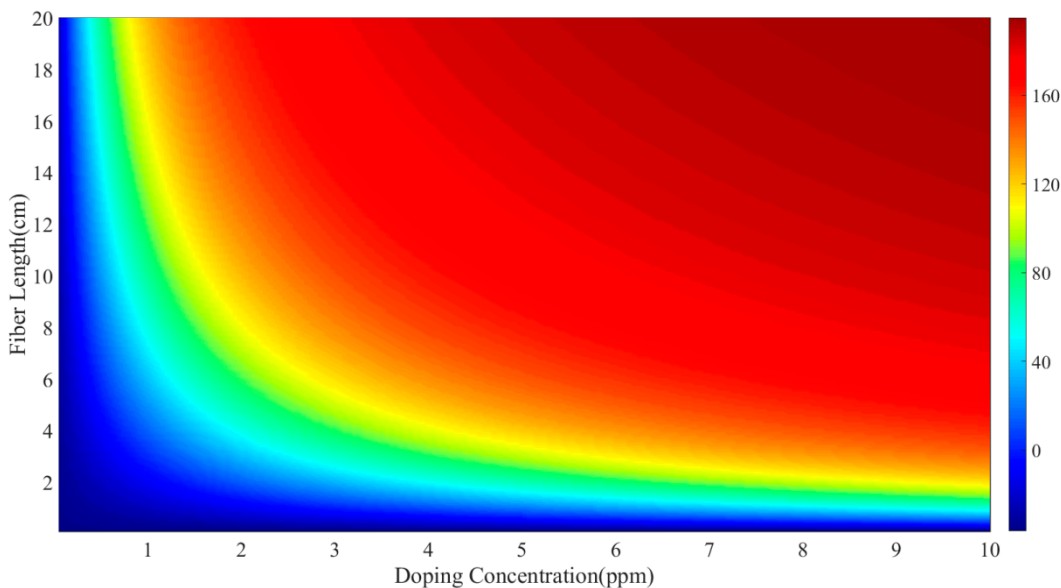

**Figure 15.** The ASE PTs of POFs with 1–20 cm lengths and 0.1 ppm–10 ppm concentrations.

### 5. Conclusions

In this work we carried out a computational study of the spectral features of light emission in CdSe/ZnS QDs-doped polymer fiber, and the evolution of the SE and ASE was analyzed quantitatively. For this purpose, we created a theoretical model based on the energy structure of the QDs and employed ad-hoc numerical algorithms in which time, distance traveled, and wavelength were independent variables. The effect of the overlap between the absorption and emission cross-sections of QDS was taken into account. The calculation results of the SE spectra based on our theoretical model were in good agreement with the experimental results, indicating the trustworthy of our model. Through analyzing the output spectra, we reproduced the red shift of the peak wavelength and average wavelength of SE and found that pump power and the overlap coefficient of absorption and emission cross-sections might be the two main factors causing the red shift of the output SE spectra. Furthermore, we found that when the pump exceeded ASE PT, FWHM will decrease rapidly with the increase of pump power, and there will be a relatively blue shift of the peak and average wavelengths. The PTs of ASE will increase with the increase of doping concentration and doped fiber length, while ASE will not be generated when the total number of QDs in POFs was less than $1.27 \times 10^{12}$. The analyses shown in this work might be helpful to design amplifiers and lasers based on QDs and POFs.

**Author Contributions:** Conceptualization, X.P. and W.L.; methodology, X.P.; software, W.L.; validation, Z.W., Y.D. and C.Y.; formal analysis, C.Y.; investigation, X.P.; resources, X.P. and W.L.; data curation, Z.W.; writing—original draft preparation, X.P.; writing—review and editing, X.P.; visualization, W.L.; supervision, W.L.; project administration, X.P. All authors have read and agreed to the published version of the manuscript.

**Funding:** This work was financially supported by the Natural Science Foundation of Zhejiang Province (LY20F040003), Natural Science Foundation of Ningbo (202003N4158), and sponsored by the K. C. Wong Magna Fund in Ningbo University.

**Data Availability Statement:** The data presented in this study are available on request from the corresponding author. The data are not publicly available due to further study.

**Conflicts of Interest:** The authors declare no conflict of interest.

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
