# Peer review of "Analysis of the Emission Features in CdSe/ZnS Quantum Dot-Doped Polymer Fibers"

_photonics, doi:10.3390/photonics10030327_

Round 1

Reviewer 1 Report

This manuscript reported a thorough computational study on the light emission spectral features of CdSe/ZnS quantum dot-doped polymer fibers. The author focused on the re-absorption phenomenon due to the overlap between the emission and absorption cross sections and employed ad-hoc numerical algorithms to figure out the reason for the red shift of spontaneous emission spectra. Also, the relationship between the pump threshold of amplified spontaneous emission and doping concentration along with doped fiber length was discussed. In my option, this is a good piece of work with some interesting information in developing the wavelength tunable QDs doped fiber amplifiers and lasers, however, various issues listed below have hindered the publication of the article in the present form.

1. In the introduction part, the authors need to add one or two sentences to compare the properties of fiber structures vs. thin film, and why fibers/QDs doped fibers earned more research interest.

2. The authors need to reference the work when stating “which can be seen in our previous work”. (page 2, theoretical model part)

3. Can authors explain why the fiber length instead of aspect ratio was taken into account as an important factor?

And how to determine the optimal fiber diameter?

4. What is the white phase/contrast in the TEM image in Figure 1?

5. The authors need to rephrase some expressions. For example, on page 7, the authors mentioned “The average wavelength dependent on wavelength and the corresponding intensity…” is misleading.

6. Error bar needs to be provided. And what is the error range of the computational wavelength?

Small points:

1. There are some errors in the manuscript, please correct them, i.e., excited sate (page 2), Therefor (page 6), etc.

2. Equations need to be well labeled, i.e. Equation (1)-(3).

Following the above considerations, I suggest a minor revision before acceptance of the manuscript.

Author Response

Dear Reviewer,

Thank you for your positive evaluations and useful suggestions on our manuscript (Photonics-2246445).

We agree with your comments and revised our manuscript accordingly. All the revised parts are marked in red in the manuscript.

Our detailed response points are attached.

Sincerely,

Xuefeng Peng on behalf of all authors.

Email: pengxuefeng@nbu.edu.cn

Reviewer 2 Report

In this paper, the authors paid special attention to the influence of re-absorption caused bythe overlap between the emission and absorption cross sections and developed a theoretical model to simulate and analyze the evolution of the spontaneous emission and Amplified Spontaneous Emission in QDs-doped polymer fibers. In addition, they developed ad hoc finite difference algorithms to solve the model based on the emission and absorption cross sections calculated from the measured absorption and PL spectra and some parameters of CdSe/ZnS QDs and fibers. Before the paper could proceed further for publication, the authors are advised to address the following issues.

Q1 It would be better if the author could provide a morphology and compositional characterization of CdSe/ZnS quantum dot doped polymer fibers. This will be a good part of the draft.

Q2 In addition to the traditional II-VI and III-V quantum dots, the new types of perovskite quantum dots should also be introduced in the Introduction (Nanoscale, 2020, 12, 6403-6410).

Q3 For fluorescent material,  photostability and thermal  stability has always been the focus of attention, so it is recommended to increase the work on stability.

Q4. In Figure 3, the abscissa should be in nanometers rather than microns. Therefore, I suggest that the author should further improve quality, including writing and figures.

Q5. It is very interesting that the emission spectra can be adjusted by adding different fiber lengths. I’d like to hear more explanations and and further proves.

Q6 The quality of the schematic shown in Figure 2 is simple and unclear. The authors should improve them before the publication of this work.

Author Response

(The authors gave the same response as above.)

Round 2

Reviewer 2 Report

I appreciate that the authors consider my opinions. The article, in its present form, is much improved with respect to its previous edition. All the questions I raised have been well addressed or explained. I recommend it for publication without further revision.